# Neutral Sphingomyelinase 2 Inhibition Limits Hepatic Steatosis and Inflammation

**DOI:** 10.3390/cells13050463

**Published:** 2024-03-06

**Authors:** Fatema Al-Rashed, Hossein Arefanian, Ashraf Al Madhoun, Fatemah Bahman, Sardar Sindhu, Halemah AlSaeed, Texy Jacob, Reeby Thomas, Areej Al-Roub, Fawaz Alzaid, MD Zubbair Malik, Rasheeba Nizam, Thangavel Alphonse Thanaraj, Fahd Al-Mulla, Yusuf A. Hannun, Rasheed Ahmad

**Affiliations:** 1Immunology & Microbiology Department, Dasman Diabetes Institute, Dasman 15462, Kuwait; hossein.arefanian@dasmaninstitute.org (H.A.); fatemah.bahman@dasmaninstitute.org (F.B.); halemah.alsaeed@dasmaninstitute.org (H.A.); texy.jacob@dasmaninstitute.org (T.J.); reeby.thomas@dasmaninstitute.org (R.T.); areej.abualroub@dasmaninstitute.org (A.A.-R.); 2Animal and Imaging Core Facilities, Dasman Diabetes Institute, Dasman 15462, Kuwait; ashraf-madhoun@dasmaninstitute.org (A.A.M.); sardar.sindhu@dasmaninstitute.org (S.S.); 3Université Paris Cité, INSERM UMR-S1151, CNRS UMR-S8253, Institut Necker Enfants Malades, F-75015 Paris, France; fawaz.alzaid@inserm.fr; 4Genetics and Bioinformatics Department, Dasman Diabetes Institute, Dasman 15462, Kuwait; mohammad.malik@dasmaninstitute.org (M.Z.M.); rasheeba.iqbal@dasmaninstitute.org (R.N.); alphonse.thangavel@dasmaninstitute.org (T.A.T.); fahd.almulla@dasmaninstitute.org (F.A.-M.); 5Stony Brook Cancer Center, Stony Brook University, Stony Brook, NY 11794, USA; yusuf.hannun@stonybrookmedicine.edu

**Keywords:** steatosis, nSmase2, *Smpd3*, inflammation, lipotoxicity, NAFLD

## Abstract

Non-alcoholic fatty liver disease (NAFLD) is manifested by hepatic steatosis, insulin resistance, hepatocyte death, and systemic inflammation. Obesity induces steatosis and chronic inflammation in the liver. However, the precise mechanism underlying hepatic steatosis in the setting of obesity remains unclear. Here, we report studies that address this question. After 14 weeks on a high-fat diet (HFD) with high sucrose, C57BL/6 mice revealed a phenotype of liver steatosis. Transcriptional profiling analysis of the liver tissues was performed using RNA sequencing (RNA-seq). Our RNA-seq data revealed 692 differentially expressed genes involved in processes of lipid metabolism, oxidative stress, immune responses, and cell proliferation. Notably, the gene encoding neutral sphingomyelinase, *SMPD3*, was predominantly upregulated in the liver tissues of the mice displaying a phenotype of steatosis. Moreover, nSMase2 activity was elevated in these tissues of the liver. Pharmacological and genetic inhibition of nSMase2 prevented intracellular lipid accumulation and TNFα-induced inflammation in in-vitro HepG2-steatosis cellular model. Furthermore, nSMase2 inhibition ameliorates oxidative damage by rescuing PPARα and preventing cell death associated with high glucose/oleic acid-induced fat accumulation in HepG2 cells. Collectively, our findings highlight the prominent role of nSMase2 in hepatic steatosis, which could serve as a potential therapeutic target for NAFLD and other hepatic steatosis-linked disorders.

## 1. Introduction

Non-alcoholic fatty liver disease (NAFLD) is the most common chronic liver disease among obese and diabetic individuals [1]. It is characterized by hepatic steatosis, caused by lipid accumulation in hepatocytes. The development of hepatic lipotoxicity is considered one of the earliest signs of NAFLD and has been linked to hepatic oxidative stress, inflammatory responses, and insulin resistance [2]. Therefore, therapeutic strategies pursuing the prevention and reduction of lipid content within the hepatocytes are the most sought-after approaches. Although several studies have been performed to understand the pathogenesis of hepatic steatosis, the clinical applications of these investigations remain limited. Alterations in the molecular mechanisms that cause NAFLD and its associated complications are only partially known.

Ceramides belong to a class of sphingolipids that are commonly found in the cell membranes. They are generated through the following three metabolic pathways: (i) de novo synthesis, (ii) sphingomyelin hydrolysis, and (iii) the salvage pathway [3]. The formation of ceramides can be induced by different stimuli, such as oxidative stress and tumor necrosis factor-α (TNF-α) [4]. The liver is a primary location for ceramide formation and contains significantly higher levels of sphingolipids, particularly ceramide and sphingomyelin (SM), as compared to other organs. Accordingly, the liver is also an organ vulnerable to sphingolipotoxicity [5].

Numerous studies have implicated ceramide dysregulation in the pathogenesis of non-alcoholic fatty liver disease (NAFLD) [6,7]. Ceramides have been shown to contribute to insulin resistance, inflammation, and lipid accumulation in the hepatocytes, key features of NAFLD [8]. Additionally, they are believed to modulate cellular and whole-body metabolism, potentially exacerbating NAFLD progression [9]. Regardless, this role of ceramides is believed to be multifaceted when it comes to hepatocyte death and liver injury. The de novo pathway of ceramide synthesis, in particular, has been extensively studied and implicated in contributing to liver damage in conditions such as NAFLD. However, our understanding of other pathways, such as the sphingomyelin pathway, and their specific roles in liver injury remains limited.

Sphingomyelin synthase (SMS) has two isoforms, SMS1 and SMS2, that metabolize ceramide into SM. SM-induced ceramide generation is catalyzed by sphingomyelinases (SMases), which can be distinguished by their pH optima [10]. Considering the fact that SM-ceramide rheostat is instrumental in regulating lipid accumulation, inflammation, and pro-apoptotic signatures of the pathogenesis of hepatic steatosis, targeting this axis may be a valuable approach for treating hepatic steatosis in humans.

In this study, RNAseq-based analysis was used to gain an in-depth understanding of how the high-fat diet (HFD) feeding in mice impacts the lipid metabolism and sphingolipid generation in the liver. Here, we present a systematic characterization of liver steatosis at the transcriptional level in HFD mice as well as decipher the molecular mechanism that drives the NAFLD-like phenotype in the HepG2 cell model.

## 2. Materials and Methods

### 2.1. Animal Studies

Ten C57BL/6 male mice, 8 weeks old and weighing 23.64 ± 2.76 gms, were purchased from the Jackson Labs (Bar Harbor, ME, USA). The mice were acclimated at the Animal Facility of Dasman Diabetes Institute (DDI), Kuwait and were maintained at a room temperature of 22 °C under a 12-h light/dark cycle. The animals had a free access to a standard laboratory chow diet and water ad libitum. All animal procedures were approved by the Animal Care and Ethics Committee (ACEC) of DDI (Project#: RAAM-2016-007), and all animal experiments followed the guidelines of Animal Research Reporting of In Vivo Experiments (ARRIVE) (https://arriveguidelines.org/, accessed on 12 March 2023).

To induce steatosis by HFD feeding, mice (*n* = 5) were fed a “Western-style diet,” which is commonly characterized by its high fat content along with added sugars such as sucrose, with 58% of the calories coming from fat (Cat#: D12331i, Research Diets Inc., Madison, WI, USA) while the control mice (*n* = 5) were fed with a standard chow diet (CD) containing 6.4% of the calories from fat (Cat#: 8664, Harlan Teklad, Madison, WI, USA). The animals were fed for a period of 14 weeks and the body weights were recorded prior to feeding, and then on a weekly basis until the end of the study. Later, the animals were euthanized, and the liver tissues were harvested, sectioned, and stored at −80 °C until use, following protocols as described elsewhere [11,12].

### 2.2. Immunohistochemical (IHC) Staining

Immunohistochemistry (IHC) staining was performed as previously described [13]. In brief, 4 μm thick, paraffin-embedded sections of liver tissues were treated with xylene and then with ethanol gradient (100%, 95%, and 75%) in water. Antigen was retrieved by heating specimens in target retrieval solution (pH 6.0; Dako, Glostrup, Denmark) for 8 min, cooled and washed in PBS, quenching the endogenous peroxidase by 3% H_2_O_2_ treatment for 30 min. After blocking (5% non-fat milk for 1 h and 1% BSA for 1 h.), samples were treated overnight at room temperature with primary antibody (rabbit polyclonal anti-F4/80 antibody, ab100790, Abcam, Cambridge, UK), washed in PBS-Tween (0.5%), and incubated for 1h with the goat anti-rabbit secondary antibody conjugated with HRP polymer chain (EnVision Kit, Dako, Glostrup, Denmark). The color was developed using DAB chromogenic substrate [10]. Finally, the slides were washed, counterstained, dehydrated, cleared, and mounted, as previously described. For analysis, digital photo-micrographs (20× magnification; Pannoramic Scan, 3DHistech, Budapest, Hungary) were used and the staining intensity was quantified using ImageJ software version 1.43m (NIH, Bethesda, MD, USA) [14].

### 2.3. Hematoxylin and Eosin (H&E) Staining and Steatosis Scoring

The sections of liver tissues (4 μm thick) were stained by hematoxylin (5 min), followed by a tap water rinse for 1 min. Eosin was applied (30 s) to counterstain and sections were re-dehydrated and set in a xylene-based mounting medium. To determine macro- and micro-vesicular steatosis scores, images were converted from RGB to a 16-bit grayscale format. Thresholds were then adjusted to the internode parameter to effectively highlight all vacuoles for accurate counting.

For further analysis, binary images of particles were adjusted for circularity and compared together with the original H&E images to ensure precision. To analyze, area percentage and mean gray value were selected, which provided information regarding the H&E image size and staining %age of the cells. The values were averaged and statistically analyzed as described previously [15].

### 2.4. Quantification of Triglyceride Levels in the Liver Tissue

Approximately 100 mg of the liver tissue was homogenized in 1 mL of 5%NP-40/ddH2O solution. Samples were heated slowly from 80–100 °C for 5 min, followed by cooling to room temperature. Samples were centrifuged at 15,000 RPM for 2 min; insoluble material was removed. Samples were then diluted 10-fold with ddH2O. The liver triglyceride levels were measured using the Triglyceride Quantification Assay Kit (Abcam, ab-65336), according to the manufacturer’s instructions.

### 2.5. RNA Sequencing and Data Analysis

RNA was extracted from tissue samples (RNeasy kit, Qiagen, Hilden, Germany), following the manufacturer’s instructions. For transcriptome libraries, 40 ng RNA sample and the TruSeq stranded mRNA kit (Illumina Inc., San Diego, CA, USA) were used, as instructed by the manufacturer. The quality and quantity of the libraries were assessed using a bioanalyzer (Agilent, Santa Clara, CA, USA) and a qubit fluorometer (ThermoFisher Scientific, Waltham, MA, USA), respectively.

Subsequently, paired-end sequencing was carried out using the Novaseq 6000 system (Illumina Inc., CA, USA), as described elsewhere [16], and the sequencing data were analyzed as stated in other reports [17,18,19]. For quantifying the reads associated with specific genes, the HTSeq-count tool was used (HTSeq version 0.9.1) [20]. Analysis of the differentially expressed genes (DEGs) was conducted (Bioconductor package edgeR, version 4.1), using default settings [21].

### 2.6. Bioinformatics Analyses

Pathway enrichment and biological process analyses were conducted using transcriptomic data from experimental and control mice. To identify the altered biochemical signaling pathways in HFD-fed mice, compared to CD-fed mice, the DEGs with *p* < 0.05 obtained from edgeR paired analysis were compared [21]. The Database for Annotation, Visualization, and Integrated Discovery (DAVID) (https://david.ncifcrf.gov/home.jsp/, accessed on 12 March 2023) was used for gene ontology enrichment of biological processes [22]. Enriched pathways were analyzed using the Kyoto Encyclopedia of Genes and Genomes (KEGG) pathway database [23], and networks visualization was performed using Cytoscape [24].

### 2.7. aSMase/nSMase2 Activity Assays

Liver tissues were homogenized in PBS with NP-40 (0.1%) and sonicated (20 W) on ice for 30 min. Tissue homogenates were kept on ice for 30 min, clarified by centrifugation (16,000× *g* for 10 min), and the activities of acidic SMase (aSMase) and neutral SMase 2 (nSMase2) were assayed. The enzyme activity was assayed in two buffer systems containing 50 µg total protein lysate and 10 µM Tris-MgCl2, at pH 5.0 for aSMase and at pH 7.4 for nSMase2, and using Amplex Red sphingomyelinase assay kit (Invitrogen, Monza, Italy), following the manufacturer’s instructions. The enzymatic activity was measured at 37 °C after 30 min, at the Ex/Em ratio of 535/587 in kinetic mode.

### 2.8. HepG2 Cell Model of Steatosis

Human hepatoma HepG2 cells were propagated in triplicate wells in 6-well plates, in Dulbecco’s Modified Eagle Medium (DMEM; Invitrogen, Grand Island, NY, USA) containing 5 mM D-glucose, 1 mM sodium pyruvate, 10 mM HEPES, 100 g/mL Normocin, 50 U/mL penicillin, and 50 g/mL streptomycin (Invitrogen, Grand Island, NY, USA). Cells were incubated at 37 °C in 90% relative humidity, under 5% CO_2_. At approximately 60% confluence, the culture medium was replaced with fresh DMEM containing 25.2 mM D-Glucose and the cells were incubated for an additional 24 h. Subsequently, the cells were treated with 100 μM of oleate acid (OA) and incubated for another 24 h. Cells in control wells were similarly maintained but, instead of OA, were treated with vehicle (0.5% BSA).

In experiments involving specific inhibitors or inducers, HepG2 cells were pre-treated with these agents, such as nSMase2 synthetic inhibitor GW4869 (10 μM; Chem Cruz, SC-218578, Dallas, TX, USA), nSMase2 inducer daunorubicin (DN) hydrochloride (1 μM; Chem Cruz, SC-200921, Dallas, TX, USA), or aSMase synthetic inhibitor imipramine (Imip) hydrochloride (10 μM; Sigma-Aldrich, 113-52-0, Burlington, MA, USA). Later on, cells were lipid-challenged (100 μM of OA) to initiate steatosis, in the presence of an inhibitor or an inducer.

### 2.9. MTT Assay

Cellular metabolic activity, as reflected by cell viability and proliferation, was assessed using MTT (3-[4,5-dimethylthiazol-2-yl]-2,5 diphenyl tetrazolium bromide) assay (Promega Bio Sciences LLC, San Luis Obispo, CA, USA), as described elsewhere [25], Briefly, 5 × 10^3^ HepG2 cells were cultured in a 96-well plate and incubated at 37 °C in 5% CO^2^ for 18 h. Cells were subjected to various conditions to study the effects of high glucose and/or oleate acid (OA) challenge on cellular metabolic activity. In some experiments, cells were pre-treated with different inhibitors and then challenged with 100 μM OA. Later, 1 mg/mL MTT reagent (a yellow tetrazole) was added to each well, the resultant purple formazan crystals were allowed to dissolve in a 10% SDS/0.04 eq/L HCl solution at 37 °C for 1 h, and the absorbance was read at 490 nm using a microplate reader (DTX880; Beckman Coulter, Brea, CA, USA). Cell survival was calculated as percentage of control cell survival using the formula: CS (%) = (mean absorbance of treated well/mean absorbance of control well) × 100.

### 2.10. Flow Cytometry

Cell apoptosis was detected using the Annexin V apoptosis detection kit and propidium iodide (PI) staining for flow cytometry (Sigma-Aldrich, St. Louis, MO, USA). HepG2 cell monolayers were trypsinized; cells were collected by centrifugation (400× *g* for 5 min), resuspended in 1× binding buffer, and stained using 5 μL Annexin V and 10 μL PI at room temperature for 15 min. Fluorescence-activated cell-sorting (FACS) analysis was carried out using a FACSCanto II flow cytometer (BD Bioscience, San Jose, CA, USA). For each acquisition, 20,000 events were recorded and analyzed. Unstained cells were used to establish the baseline for the negative and positive gates in the analysis, and BD FACSDiva™ Software 8 (BD Biosciences, San Jose, CA, USA) was used for data analysis.

### 2.11. BODIPY Lipid Staining and Quantification

To quantify lipids in HepG2 cells, BODIPY 495/503 staining (Cat#: D3922, Life Technologies, Carlsbad, CA, USA, Invitrogen) was used. Briefly, cells were washed using FACS buffer and resuspended in 2% paraformaldehyde (PFA) solution. After centrifugation, cells were stained for 15 min at room temperature with BODIPY 493/503 using a concentration of 2 μM, resuspended in FACS buffer, and analyzed using a FACSCanto II flow cytometer (BD Bioscience, San Jose, CA, USA). FACSDiva Software 8 (BD Biosciences, San Jose, CA, USA) was used for data analysis. Staining Index (SI) was calculated based on mean fluorescence intensity (MFI) difference between positive and negative populations, divided by two times the standard deviation of the negative (unstained) population.

### 2.12. Confocal Microscopy

To assess intracellular lipid accumulation, HepG2 cells seeded on coverslips were exposed to stimuli for 4 h, fixed by using 4% PFA for 10 min, and stained with Nile Red for 15 min. After washing three times with PBS, 5 min each, specimens were counterstained and mounted in DAPI Vectashield medium (Vectorlab, H1500, Newark, CA, USA). Images were captured using a Zeiss LSM710 spectral confocal microscope (Carl Zeiss, Gottingen, Germany). Excitation at 543 nm (HeNe laser) and 405 nm (argon ion laser), and emissions were detected and analyzed. Correlated Total Cell Fluorescence (CTCF) was calculated as follows:CTCF = Integrated Density (IntDen) − (Area of selected cells × Background Mean Grey Value (BMGV)).

### 2.13. Real-Time RT-qPCR

Total RNA was extracted from liver tissues or HepG2 cells (RNeasy kit, Qiagen, Valencia, CA, USA), following the manufacturer’s protocol. cDNA was synthesized from 0.5 µg RNA template (High-Capacity cDNA Reverse Transcription kit, Applied Biosystems, Waltham, MA, USA). Real-time RT-qPCR was carried out as previously described [26]. Briefly, 50 ng cDNA sample and TaqMan Gene Expression Master Mix (Applied Biosystems, Foster City, CA, USA) were used with gene-specific TaqMan assays for following genes: Tumor necrosis factor-alpha (*TNF-α*): Assay Hs01113624_g1; Peroxisome proliferators activated receptors-alpha (*PPARα*): Assay Hs00947536_m1; Cytochrome P450 2E1 (*CYP2E1*): Assay ID: HS00559367_M1; Acetyl-CoA Carboxylase Alpha (*ACACA*): Assay ID: Hs01046047_m1; Fatty acid synthase (*FASN*): Assay ID: Hs01005622_m1; Carnitine palmitoyltransferase 1A (*CPT1α*): Assay ID: Hs00912671_m1; Carnitine palmitoyltransferase 2 (*CPT2*): Assay ID: Hs00988962_m1; Interleukin 1 Beta (*IL-1β*): Assay ID: Hs01555410_m1; Interleukin 6 (IL-6): Assay ID: Hs00174131_m1; DNA damage-inducible transcript 3 (*DDIT3*): Assay ID: Hs00358796_g1; Sterol Regulatory Element Binding Transcription Factor 1 (*SREBF1*): Assay ID: HS01088691_m1; Glyceraldehyde 3-phosphate dehydrogenase (*GAPDH*): Assay ID: Hs03929097_g1, Sphingomyelin phosphodiesterase 1 (*SMPD1*): Assay ID: Hs03679346_g1/Mm00488318_m1, Sphingomyelin synthase 1 (SGMS1): Assay ID: HS00983630_m1, sphingomyelin synthase 2 (*SGMS2*): Assay ID: Hs00380453_m1, Sphingomyelin phosphodiesterase 3 (*SMPD3*): Assay ID: Hs00920354_m/Mm00491359_m1), along with target-specific TaqMan MGB probes labeled with FAM at the 5′-end and NFQ-MGB at the 3′-end of the probe, using a 7500 Fast Real-Time PCR System (Applied Biosystems, Foster City, CA, USA). PCR cycles used were as follows: 15 s at 95 °C for denaturation, 2 min at 50 °C for uracil-DNA glycosylases activity, 1 min at 60 °C for annealing/extension, and 10 min at 95 °C for AmpliTaq gold enzyme activation. Gene expression relative to that of control was determined using the comparative CT method, and normalized to GAPDH expression. Results (mean ± SEM) were expressed as fold change relative to control [27].

### 2.14. Western Blotting

Target protein expression was determined by Western blotting as described elsewhere. Briefly, HepG2 cells were lysed in RIPA buffer, and extracted proteins were quantified using Quick Start Bradford assay (Bio-Rad, Hercules, CA, USA). Equal amounts of protein samples were resolved on 12% polyacrylamide gels, transferred to nitrocellulose membranes (Bio-Rad, Hercules, CA, USA), blocked and treated with primary Abs against TNF-α (Cell Signaling Technology; Cat#: 6945, Danvers, MA, USA), PPARα (Abcam; Cat#: ab3484, Cambridge, UK), or β-actin (Cell Signaling Technology; Cat#: 3700T, Danvers, MA, USA), followed by treatment with HRP-linked secondary Ab (Promega, Madison, WI, USA). Protein bands were developed using the SuperSignal West Femto ECL kit (Thermo Scientific, Waltham, MA, USA), and images were captured (ChemiDoc MP imaging system, Bio-Rad, Hercules, CA, USA).

### 2.15. Enzyme-Linked Immunosorbent Assay (ELISA)

TNF-α levels in HepG2 culture supernatants were determined using commercial sandwich ELISA (R&D systems, Minneapolis, MN, USA), following the recommended protocol by the manufacturer, and as described [28].

### 2.16. Statistical Analyses

Statistical analyses were performed using GraphPad Prism (La Jolla, San Diego, CA, USA) and SPSS for Windows version 19.01 (IBM SPSS Inc., Armonk, NY, USA) [29]. Data are presented as mean ± SEM. Unpaired Student’s t-test was used to compare means vs. control. Pearson correlation was used to determine associations between variables. *p*-value ≤ 0.05 was considered significant.

## 3. Results

### 3.1. Chronic HFD Feeding Induces NAFLD Phenotype in Mice

Regarding the role of chronic HFD feeding in causing hepatic injury and NAFLD in mice, we found that a 14-week HFD feeding in mice led to significant weight gains, compared to control mice fed with CD (Figure 1A). The livers from HFD-fed mice had a pale discoloration with greasy consistency and were significantly enlarged (*p* = 0.05), compared to those from CD-fed mice (Figure 1B,C). At histopathological examination, both macro- and microvesicular steatosis with lobular inflammation were observed in the livers of HFD-fed mice, compared to CD-fed mice (Figure 1D,E). As expected, the livers of HFD-fed mice also displayed increased macrophage infiltration (Figure 1F,G; *p* ˂ 0.001) and lipid accumulation (Figure 1H–J; *p* ˂ 0.01), compared to the livers of CD-fed mice. However, no significant impact was observed on both Aspartate Aminotransferase Activity (AST) nor Alanine Aminotransferase Activity (ALT) Assays (Appendix A).

### 3.2. Transcriptomic/Bioinformatic Analyses Reveal Smpd3 (nSMase2) as a Potential Candidate Gene Associated with Liver Steatosis in HFD Mice

The RNA-seq approach was used to identify potential gene target(s) in HFD mice with hepatic steatosis. Principal component analysis (PCA) revealed differential clustering patterns between the two dietary interventions (Figure 2A). To show the differentially expressed genes (DEGs) in the livers of HFD mice, significantly altered genes were mapped in a volcano plot (Figure 2B). Overall, 510 genes were upregulated and 672 genes were downregulated in HFD group, compared to CD group. Next, the GO enrichment analysis further revealed that, in steatotic livers, the upregulated genes were related to lipid metabolism, oxidative stress, immune responses, cell proliferation, and apoptosis (Figure 2C), while the downregulated genes were related to triglyceride biosynthesis, gluconeogenesis, glucose response, inflammatory and the endoplasmic reticulum (ER) responses (Figure 2D). KEGG pathway analysis showed that the upregulated genes were associated, per previous literature reports, with cholesterol metabolism, MAPK signaling, steroid biosynthetic processes, NF-κB and Wnt signaling, and bile secretion (Figure 2E). The downregulated genes were related to pathways of metabolism, AMPK signaling, fatty acid degradation, PPAR and PI3K-Akt signaling, insulin resistance, and alcoholic liver disease (Figure 2F). The most significantly enriched DEGs were associated with altered functions related to cholesterol metabolism, reactive oxygen species, apoptosis, response to ATP, lipid metabolism, and fatty acid metabolism (Figure 2G). Further, Chord plot of the relationship between genes and top significant GO terms showed that steatotic livers had increased expression of genes involved in lipid and fatty acid metabolism. Remarkably, Smpd3 was identified as one of such highly upregulated genes. It encodes for nSMase2, an enzyme which catalyzes the hydrolysis of sphingomyelin into ceramide and phosphocholine [30,31] (Figure 2H). Different chronic liver conditions were associated with changes in SM levels due to altered SMases [32]. To further verify the RNA-Seq data, *Smpd3*/nSMase2 and Smpd1/aSMase gene expression was confirmed by RT-qPCR. The data show a significant upregulation of *Smpd3*/nSMase2 in the livers of HFD mice, whereas no significant changes were observed in the expression of aSMase, compared to CD mice (Appendix A). As expected, a significant elevation in nSMase2 activity was detected in the livers of HFD mice, compared to CD mice (Figure 2I). Collectively, these data indicate a dysregulation in sphingolipid metabolism, from upregulated Smpd3 expression and increased nSMase 2 activity in steatotic livers of HFD mice.

### 3.3. Characterization of HepG2 Cell Model of Steatosis and Role of SM Pathway 

To further understand the role of SM pathway, the HepG2 cell model mimicking the characteristics of steatosis was established (Appendix A). In this cell model, as expected, a significant intracellular lipid accumulation was observed, as indicated by both Nile red staining and BODIPY flow cytometry (*p* ≤ 0.05; Appendix A). Given that the progression of steatosis is paralleled by upregulation of TNF-α and other inflammatory markers [33,34], we measured TNF-α production in the HepG2 cell model. To this end, TNF-α was found to be elevated in response to high glucose and OA treatments (*p* ≤ 0.001), compared to the corresponding control treatments (Appendix A).

Under high glucose and high free fatty acid conditions, both the lipid accumulation and inflammation are triggered in HepG2 cells, most likely mediated by induction of oxidative stress [35]. To test this, the expression of PPARα, which is a nuclear modulator known to protect against oxidative damage [36,37], was determined. To this end, a significant reduction in *PPARα* gene expression (*p* ≤ 0.001) was observed in HepG2 cells after challenge with high glucose or with OA (Appendix A). In concordance, a significant reduction in PPARα protein expression was also noted (Appendix A), indicating a compromised cellular protection against oxidative stress. On the other hand, the gene expression of cytochrome P450 2E1 (*CYP2E1*), a contributor to oxidative stress in NAFLD [36], was found to be elevated in HepG2 cell model of steatosis (Appendix A).

Given that hepatocyte apoptosis is a hallmark of NAFLD, we determined whether HepG2 cell challenge by high glucose or OA-induced apoptosis. To this effect, annexin V/PI staining revealed apoptosis in HepG2 cells, compared to control (Appendix A). Accordingly, MFI of annexin V-FITC A staining in apoptotic cells was found to be significantly high (*p* ≤ 0.001), compared to controls (Appendix A). High glucose and OA treatments led to a significant reduction in cell survival (*p* ≤ 0.00001; Appendix A). Collectively, these data suggest that induction of a glucolipotoxic stress triggers inflammation, oxidative stress, and apoptosis in the HepG2 cell model—the features that mark liver steatosis in HFD mice or humans with NAFLD.

Next, changes in the expression of the genes were determined that are known to be involved in SM pathway (Figure 3A). To this end, gene expression of sphingomyelin synthase 1 (*SGMS1*) (Figure 3B) and *SGMS2* (Figure 3C) was found to be downregulated in HepG2 cells (*p* ≤ 0.05), suggesting a suppression in SM synthesis. On the other hand, regarding expression of genes regulating SM hydrolysis, only the *Smpd3*/nSMase2 expression was increased in the challenged cells (*p* ˂ 0.01), while SMPD1/aSMase expression differed non-significantly from control (Figure 3D,E).

### 3.4. Effect of nSMase2 Inhibition in HepG2 Cell Model of Steatosis

To further evaluate the effect of the inhibition of nSMase2 in the HepG2 cell model, both the loss- and gain-of-function approaches were tested by using nSMase2 synthetic inhibitor GW4869 and nSMase2 agonist Daunorubicin (DNR), respectively. We also tested the effect of the inhibition of aSMase by using its functional inhibitor Imipramine (Imip). The nSMase2 activity was significantly increased (*p* ≤ 0.0001) under steatosis conditions (high glucose/OA), compared to normal/basal conditions. Likewise, nSMase2 activity was significantly elevated in response to DNR treatment. However, Imip treatment had no effect on the basal nSMase2 activity, and GW4869 inhibited the basal nSMase2 activity (Figure 4A). As anticipated, aSMase enzymatic activity was significantly reduced by treatment with Imip, while no significant effect was noted by treatments with GW4869 or DNR (Figure 4B).

To investigate the functional effect of the inhibition of nSMase2 activity in the HepG2 cell model of steatosis, cells were cultured under high glucose conditions and treated with GW4869, DNR, or Imip, prior to challenge with OA to induce steatosis. In this regard, comparing to the lipid accumulation under steatosis-inducing conditions, pre-treatment with GW4869 or Imip resulted in a significant reduction in the total lipid content in HepG2 cells treated with OA (Figure 4C–E)).

To further elucidate the impact of our treatments on model lipid metabolism, gene expression analysis for lipogenesis genes was conducted. A discernible trend in Acetyl-CoA carboxylase alpha (*ACACA*) gene expression was observed in HepG2 cells treated with OA, with a slight elevation following GW4869 and Imip treatment (Figure 4F). However, this observation did not attain statistical significance. Notably, downstream of *ACACA*, the gene expression of Fatty Acid Synthase (*FASN*) corroborated this observation, demonstrating a significant reduction in lipogenesis in our steatosis model, and a significant elevation in lipogenesis under both nSMase and aSMase inhibition (Figure 4G). This dataset further bolsters our observation of lipid accumulation, as lipogenesis is attenuated amidst lipid accumulation, reflecting homeostatic mechanisms at play. In a parallel manner, we endeavored to investigate the impact on beta-oxidation genes. Our steatosis model exhibited no significant impact on either Carnitine palmitoyltransferase 1a (*CPT1a*) or Carnitine palmitoyltransferase 2 (*CPT2*) expression. However, intriguingly, it was observed that both Imip and DNR treatments induced a significant upregulation in *CPT1a* expression with no significant impact on *CPT2* (Figure 4H, I). This phenomenon may be elucidated by considering the mechanism of action of these inhibitors, as they could potentially alleviate the cellular stress associated with lipid accumulation, thereby facilitating a compensatory response aimed at enhancing fatty acid oxidation pathways, such as beta-oxidation, through upregulation of *CPT1a*. This adaptive response may represent an attempt by the cells to mitigate lipid overload and maintain metabolic homeostasis.

### 3.5. nSMase2 Inhibition Reduces Inflammation, Oxidative Stress, and Apoptosis in HepG2 Cells

The SM pathway-derived ceramide is associated with inflammatory signaling and both nSMase2 and aSMase were found to induce expression of inflammatory cytokines/chemokines, including TNF-α and other proinflammatory proteins [38,39]. Regarding whether the nSMase2 inhibition could suppress inflammatory response in the HepG2 cell model of steatosis, the data show that nSMase2 inhibition using GW4869 significantly downregulated *TNF-α* expression when the cells were challenged with steatosis-inducing stimuli (*p* ≤ 0.01); however, *TNF-α* suppression using aSMase inhibitor Imip did not reach the statistical significance (Figure 5A). Indeed, the treatment with either inhibitor resulted in a significant suppression in TNF-α protein (*p* ≤ 0.01) (Figure 5B). Notably, a similar reduction in TNF-α secretory protein was observed (Figure 5C) Moreover, this observation extended to other tested cytokines’ gene expression, including *IL-1b* and *IL-6*, where similar trends were observed. Both GW4869 and Imip treatments resulted in a significant suppression of *IL-1b* and *IL-6* gene expression, reinforcing the anti-inflammatory effects of inhibiting sphingomyelinase pathways in the context of hepatic steatosis. Notably, while Imip treatment showed a reduction in *IL-6* expression, this reduction did not reach statistical significance (Figure 5D,E).

Furthermore, nSMase2 inhibition by GW4869 led to a significant increase in PPARα gene and protein expression, whereas aSMase inhibition by Imip had a non-significant effect on *PPARα* gene expression but led to a significant reduction in PPARα protein expression (*p* < 0.01). As expected, stimulation with nSMase2 agonist DNR led to a significant reduction in PPARα gene/protein expression (*p* ≤ 0.01; Figure 5F,H). Furthermore, a pattern of changes similar to TNF-α expression was noted for CYP2E1 expression, indicating an increase in ER stress (Figure 5I). To further corroborate these findings, we evaluated additional markers of ER stress, including DNA Damage Inducible Transcript 3 (*DDIT3*) and Sterol Regulatory Element-Binding Proteins (*SREBP*). Consistently, an increase in *DDIT3* gene expression was observed in our liver steatosis model, which diminished with GW4869 treatment (Figure 5J). However, *SREBP* gene expression did not show a significant difference between the liver steatosis model and the normal condition. Nevertheless, the suppression of aSMase via Imip notably elevated *SREBP* expression compared to the normal condition, though this increase was not significantly different from that observed in liver steatosis (Figure 5K). This suggests that aSMase inhibition may activate ER stress through an alternative pathway.

As expected, nSMase 2 inhibition by GW4869 significantly improved the viability of HepG2 cells, compared to treatments with other inhibitors (Figure 5L and Appendix A). Together, these findings suggest that nSMase2 inhibition reduces inflammation, oxidative stress, and apoptosis in HepG2 cells.

### 3.6. nSMase2 Deficiency Improves Steatosis-Associated Pathological Changes in HepG2 Cells 

Next, we asked if the nSMase2 deficiency was sufficient to improve steatosis-associated changes in the HepG2 cells model. To this end, genetic suppression of nSMase2/*Smpd3* by siRNA rendered a moderate level ~30–40% transcriptional repression, compared to control (expression in scrambled siRNA-transfected cells) (Figure 6A). Notably, even at this moderate level of suppression, the nSMase2 deficiency caused a significant reduction in TNF-α transcripts (Figure 6B) and protein expression (Figure 6C) in HepG2 cells under steatosis-inducing conditions, implying that nSMase2 had a potential role in inducing inflammation in this model of steatosis. As expected, genetic suppression of nSMase2/*Smpd3* also improved oxidative stress, as indicated by upregulation of PPARα gene (Figure 6D) and protein (Figure 6E) expression as well as by transcriptional downregulation of CYP2E1 (Figure 6F). In addition, nSMase2 deficiency also led to a significant reduction in intracellular lipid accumulation (Figure 6G) and improved cell survivability (Figure 6H–J). Overall, these results highlight the potential role of nSMase2 in driving steatosis-related pathological changes in the HepG2 cell model.

## 4. Discussion

Despite the intensive ongoing research focusing on preventing or restraining the progression of obesity-induced hepatic steatosis, more effective options remain limited. One of the main hurdles in drug discovery studies targeting NAFLD development is the absence of a reliable in vitro human model. Although murine models have been widely used to identify metabolic pathways that are associated with NAFLD, translating the preclinical findings from murine models to humans can be challenging in certain cases. In this study, RNAseq analysis identified changes in the liver at the transcriptional level that were induced or promoted by a 14-week HFD feeding in mice. The functional analyses of the identified DEGs revealed an association with a multitude of pathobiological processes including lipid metabolism, oxidative stress, immune response, cell proliferation, and apoptosis. More importantly, RNAseq data identified the involvement of Smpd3 (encoding for nSMAse2) as a key driver of hepatic changes in HFD-fed mice. The upregulated Smpd3 expression was supported by increased nSMAse2 activity in the livers of HFD-fed mice. Sphingolipids are implicated in a wide variety of biological functions such as inflammation, immune response, cell cycle, adhesion, and migration [40,41]. Ceramide, a core product of sphingolipid metabolism, is known to play a role in hepatocellular apoptosis and fibrosis [3,42]. Recently, Wang et al. highlighted the role of RNA methyltransferases, Mettl3 and Mettl14, by identifying that Mettl3 deficiency led to a decreased decay of its downstream target Smpd3 and disruptions in sphingolipid metabolism. Consequently, the accumulation of toxic ceramides caused damage to mitochondrial function and ER stress, leading to aberrant growth and function maturation during postnatal liver development in male mice [32]. Thus, this study underscores the critical role of Smpd3 in normal liver development and function. In extrapolation of their work, we also observed a critical role of Smpd3 (increased nSMase 2 activity), albeit in inducing NAFLD in HFD-fed mice. Increased lipid storage in the liver caused by excessive energy consumption may promote hepatic inflammation and oxidative stress, and result in several metabolic disorders such as hypertension, diabetes, and NAFLD [43,44].

To further understand the molecular mechanism of how HFD feeding induced hepatic steatosis and favored ceramide generation via the SM pathway, we developed a HepG2 cell-based hepatic steatosis/NAFLD model. This model pivoted on consistent stress induction in these cells by a shift over from normal (5.6 mmol) to high (25.2 mmol) glucose concentration in the culture medium, followed by a fatty acid-mediated lipotoxic stress using oleic acid. Of note, levels of intracellular lipids increased when the cellular metabolism was switched from low glucose/low fat to a high abundance of both. This phenomenon has previously been demonstrated in mice lacking thyroid hormone receptor-α, where the whole-body energy metabolism changes affected the hepatic lipid (diacylglycerol) content and rates of hepatic glucose and fatty acid oxidation, leading to improved hepatic and peripheral insulin sensitivity [45]. Several studies support that diverse glucose fluctuations might be a causal player in triggering diabetic complications [46,47,48]. Interestingly, we showed earlier that glucose fluctuations from low to high could trigger pro-inflammatory macrophage responses [48]. Indeed, inducing a similar type of glucotoxic stress in HepG2 cells also triggered an inflammatory response and promoted the expression of a prototypic proinflammatory cytokine TNF-α in this HepG2 cell model. Induction of a metabolic challenge from high glucose/high fatty acid levels led to recapitulation of NAFLD-like pathologies in the HepG2 cell model via the signatures of steatosis, oxidative stress, and cellular apoptosis. Such pathophysiological alterations in the liver are observed in patients with type 2 diabetes or those with manifestation of metabolic syndromes, such as NAFLD [49,50,51,52], overall showing a significant resemblance with the phenotypes observed in our HepG2 cell model of steatosis.

In our cell model of steatosis, we found that transcripts’ levels of Smsg11 and Smsg2 were significantly downregulated, indicating a reduction in SM synthesis. A similar effect was previously observed in the Li et al. study, showing that Sms1/Sms2 double knockout female mice fed on fat developed higher liver triglycerides and cholesterol levels [53]. In our cells model, we also found a significant upregulation of Smpd3 expression and nSMase2 activity under steatosis-promoting conditions, similar to the increased nSMase2 activity in the liver tissues of HFD-fed mice. During the onset of diabetes, the circulatory levels of several pro-inflammatory cytokines, including TNF-α, are elevated. We and others previously showed that TNF-α upregulation can initiate ceramide synthesis by binding to TNFR1 and activating the nSMases2 activity that hydrolyzes SM to ceramide [38,54]. Of note, in our HepG2 cell model, the nSMase2 loss- or gain-of-function studies revealed that the inhibition of this enzyme significantly prevented the accumulation of intracellular lipids, while on the other hand its agonist-associated overactivation promoted the intracellular lipid accumulation. Additionally, the indirect inhibition of aSMase also showed a significant reduction in the intracellular lipid accumulation. Inhibition of both nSMase2 and aSMase also reduced inflammatory response induced by gluco-lipotoxic stresses. This observation is concordant with other studies that established the link between inflammation and the degradation of SMase activity for converting SM into ceramide [55,56,57,58].

Oxidative stress is a major factor that leads to inflammation, especially during aging [59]. PPARα activation is known to play a protective role regarding both fatty acid-induced oxidative stress and apoptosis [60,61,62]. Montagner A. et al. showed that hepatocyte PPARα deletion led to an impaired fatty acid homeostasis in the liver and hepatic lipid buildup in PPARα hep−/− mouse model [63]. Indeed, in our HepG2 cell model of steatosis, the lipotoxic effect of OA stimulation was found to be associated with downregulation of PPARα and upregulation of CYP2E1 proteins. Remarkably, the inhibition or deficiency of nSMase2 increased the PPARα activity, reduced the CYP2E1 expression, and favored cell survival. This interesting observation was limited to nSMase2 and was not observed for aSMase. However, it is important to note that Imip is a well-known aSMase inhibitor, but its impact is indirect by disrupting the intracellular lysosomal membrane and inhibiting aSMase binding to the membrane, a process that results in aSMase degradation and ultimately its deactivation. For this reason, it may be interesting to further study the plausible effects of aSMase inhibition by gene silencing or knock-down in the HepG2 cell model of steatosis that was used.

The strength of this work lies in its comprehensive approach to understanding obesity-induced hepatic steatosis and NAFLD development. The study effectively bridges the gap between murine models and human cellular models, providing valuable insights into the molecular mechanisms underlying these conditions. The use of RNAseq analysis to identify transcriptional changes in liver tissues of HFD-fed mice is particularly notable, as it highlights the potential involvement of nSMase2/Smpd3 in hepatic changes. This is further supported by functional analyses linking these changes to various pathobiological processes such as lipid metabolism and oxidative stress. Additionally, the development of a HepG2 cell-based model of hepatic steatosis/NAFLD under glucolipotoxic stress conditions is a significant advancement, offering a more relevant and controlled environment to study progression of the disease under the impact of novel therapeutic drug interventions. The HepG2 cell model offers a promising platform for screening new candidates and understanding their mechanisms of action in a well-controlled, human-relevant system. This approach bridges the gap between basic research and its clinical application, paving the way for more effective therapies for NAFLD and related metabolic disorders.

However, this study is also limited by certain caveats. While the HepG2 model successfully mimics several aspects of liver steatosis, it is important to remember that cell lines may not fully replicate complexity of liver tissues in vivo. This could limit the generalizability of the findings to actual human liver conditions. Furthermore, this study basically focuses on the role of nSMase2/Smpd3, thus overlooking other critical pathways and factors that could be involved in hepatic steatosis. Future research should aim to explore these areas, possibly incorporating more diverse models or methods to provide a more holistic understanding of the disease. Additionally, while the study makes significant efforts in highlighting the molecular mechanism of hepatic steatosis, translating these findings into effective clinical interventions remains a potential challenge, requiring further research and validation in clinical settings.

## 5. Conclusions

In conclusion, transcriptomic analysis of the livers from HFD-fed mice with liver steatosis identified a significant upregulation in Smpd3 (nSMase2), a key finding that led to the development of a HepG2 human cell model of induced steatosis by using glucolipotoxic stress conditions. This cell line model adeptly mimics the hallmark features of the NAFLD phenotype, demonstrating how the upregulated Smpd3/nSMase2 expression/activity could contribute to hepatic lipid accumulation, inflammation, oxidative stress, and apoptosis. Clinically, these findings hold a crucial relevance, suggesting that Smpd3/nSMase2 and the resultant biochemical pathways driving NAFLD present as potential targets for therapeutic interventions. These studies may offer further opportunities for developing new drugs or strategies aimed at modulating Smpd3/nSMase2 expression and activity to mitigate or reverse the progression of NAFLD.

## Figures and Tables

**Figure 1 cells-13-00463-f001:**
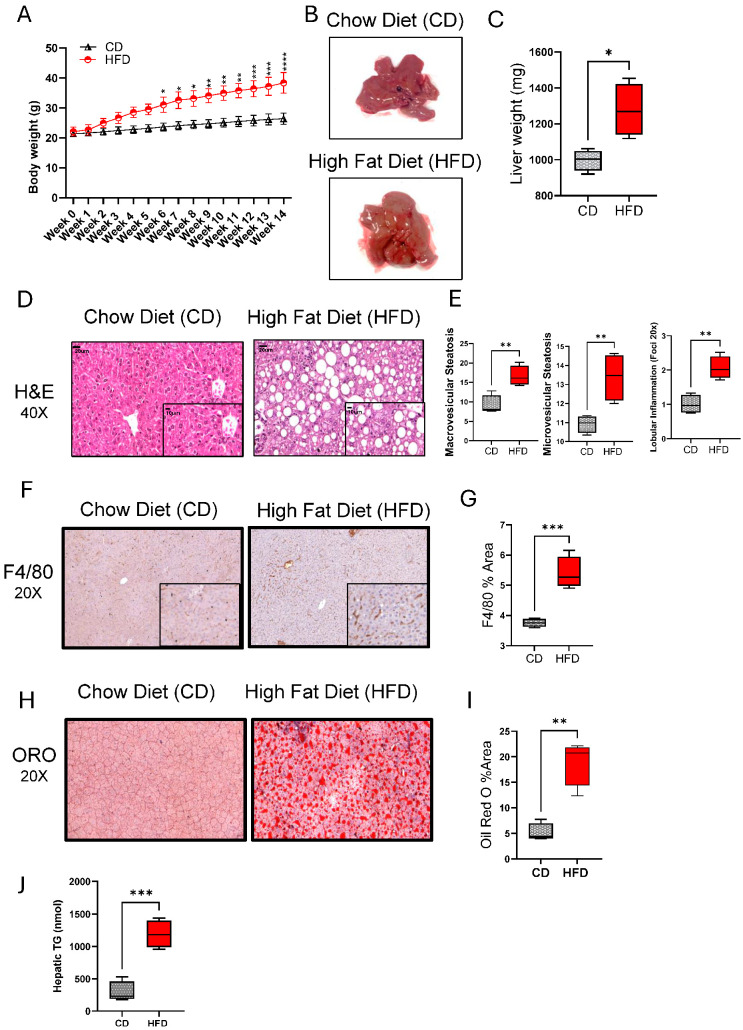
High-fat diet (HFD) feeding induces chronic liver injury and a NAFLD phenotype in mice. C57BL/6 male mice were fed HFD or chow diet (CD), 5 each, for 14 weeks, and the livers were collected after euthanasia at the end of feeding. (**A**) Body weights (g) over time are shown for HFD and CD groups. (**B**) Gross appearance of the representative livers from HFD- and CD-fed mice. (**C**) Liver weights compared between CD and HFD groups at the end of 14 week feeding. (**D**) Representative H&E stainings showing histopathological changes in the livers of HFD and CD mice. (**E**) Macrovascular and microvascular steatosis and lobular inflammation scores related to the livers of CD and HFD mice. (**F**) Representative immunohistochemical (IHC) stainings showing macrophage infiltration in the livers of HFD and CD mice. (**G**) F4/80 staining (% area) related to the livers of HFD and CD mice. (**H**) Representative Oil Red O (ORO) Staining showing intracellular lipid accumulation (**I**) ORO staining (%area) related to the liver of HFD and CD mice. (**J**) Quantification of triglyceride levels in the liver tissue. For histopathological analysis, 10 random fields/specimen were counted and viewed at 20× or 40× magnification. All data are expressed as mean ± SEM. * *p* ≤ 0.5, ** *p* ≤ 0.1, *** *p* ≤ 0.001.

**Figure 2 cells-13-00463-f002:**
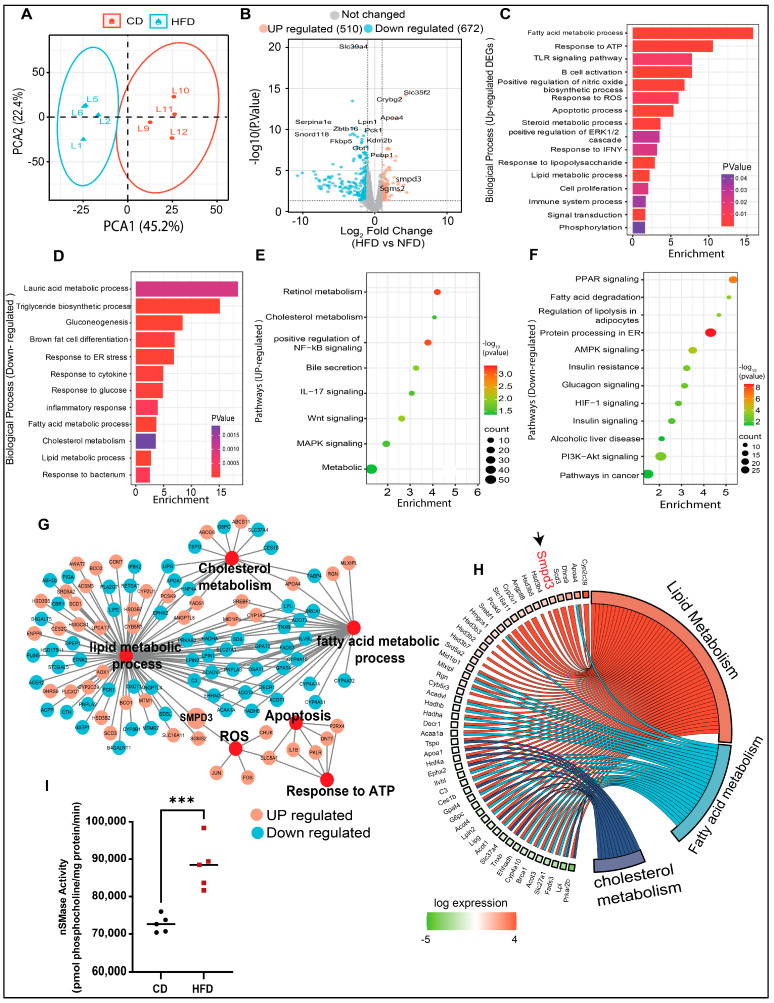
Transcriptomic and bioinformatic analyses identify the enhanced Smpd3 expression in the steatotic livers of high-fat diet (HFD)-fed mice. RNA-seq was used to identify potential gene target(s) in the livers of mice fed with a HFD, compared to those fed with a chow diet (CD), 5 mice per group. (**A**) Principal component analysis (PCA) shows differential clustering patterns in the livers of HFD and CD mice. (**B**) Volcano plot distribution highlights the differentially expressed genes (DEGs) in the livers of HFD and CD mice. Orange and cyan color dots denote the up- (510) and downregulated (672) significantly DEGs, respectively, while the gray color dots represent non-significant DEGs between CD- and HFD-fed mice. (**C**,**D**) Gene Ontology (GO) enrichment analysis based bar plots represent the biological processes that are significantly altered in relation to up- and downregulated DEGs in the liver samples from CD- and HFD-fed mice. X-axes represent the Fold Enrichment, with statistical significance indicated by a numerical value and a color code as shown. (**E**,**F**) KEGG pathway analysis of the significantly up- and down-regulated DEGs respectively, in the livers of HFD and CD mice. Dot size indicates the count which represents the number of genes associated with each pathway. Dot color denotes the statistical significance (red being the highest), and the x-axis represents the fold enrichment. (**G**) Major gene networks interconnectivity including lipid, cholesterol, and fatty acid metabolic processes, apoptosis, oxidative stress by ROS, and ATP response. Orange and cyan color dots represent the up- and downregulated, respectively, DEGs in the livers of HFD and CD mice. (**H**) Chord plot showing the relationship between genes and top significant GO terms including cholesterol, lipid, and fatty acid metabolism. Genes involved in the enrichment are arranged (on the left) in order of their expression level (−log of fold change). (**I**) Increased nSMase2 activity level in in the livers of HFD mice, compared to CD mice. *** *p* ˂ 0.001.

**Figure 3 cells-13-00463-f003:**
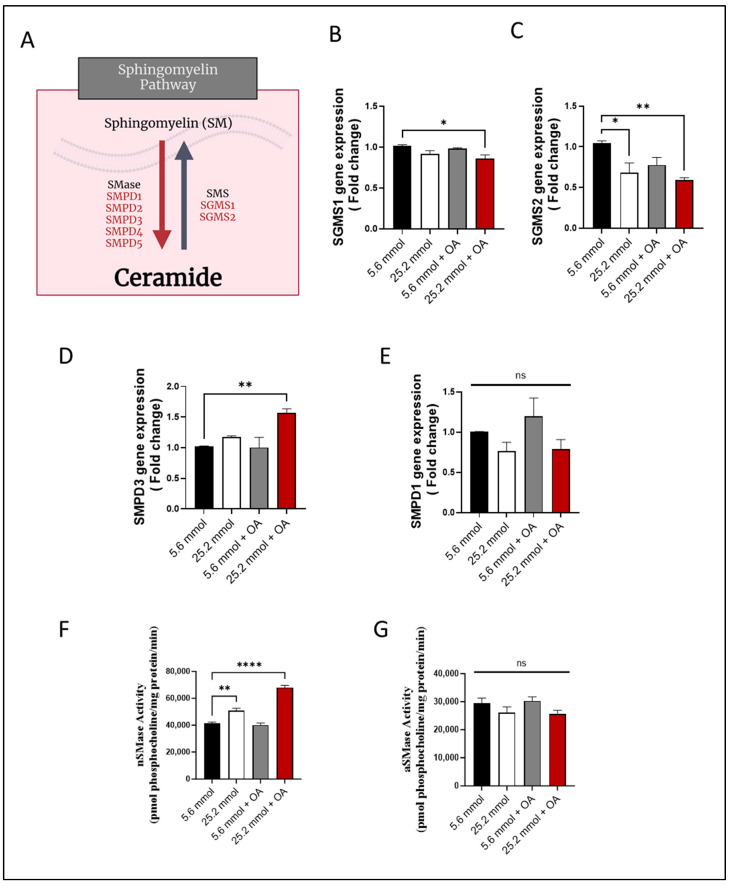
Increased Smpd3 expression and nSMase2 activity in HepG2 cell model of steatosis. HepG2 cells were cultured in normoglycemic (5.6 mmol D-glucose) or hyperglycemic (25.2 mmol D-glucose) conditions, in the presence or absence of oleic acid (OA) stimulation. Gene expression of *SGMS1* and *SGMS2* was determined using RT-qPCR; nSMase2 and aSMase activities were determined using commercial kits as described in Materials and Methods. (**A**) Schematic representation of the main sphingomyelin (SM) pathway. Reduced gene expression of (**B**) *SGMS1* and (**C**) *SGMS2* in HepG2 cells under glucolipotoxic conditions. (**D**) Increased Smpd3 gene expression in HepG2 cells under glucolipotoxic conditions. (**E**) *Smpd1* gene expression in HepG2 cells differed non-significantly between glucolipotoxic and normal conditions. (**F**) Increased nSMase activity in HepG2 cells under glucolipotoxic conditions. (**G**) aSMase activity in HepG2 cells differed non-significantly between glucolipotoxic and normal conditions. All data are expressed as mean ± SEM. * *p* ≤ 0.05, ** *p* ≤ 0.01, **** *p* ≤ 0.0001, ns: non-significant.

**Figure 4 cells-13-00463-f004:**
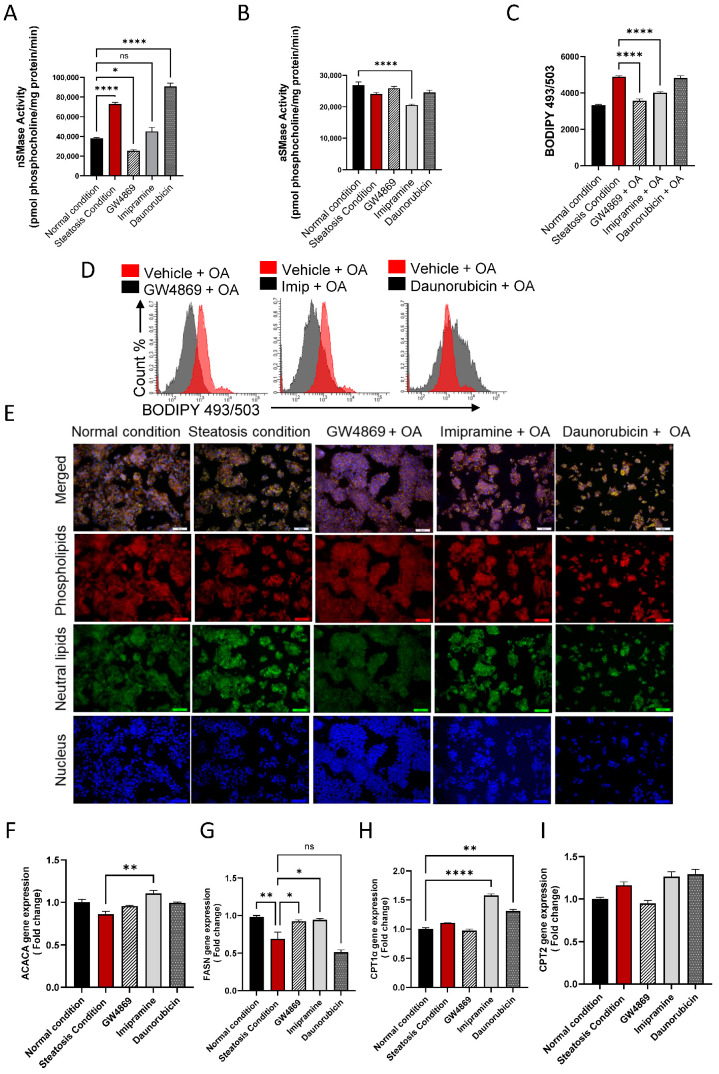
Effect of nSMase2 inhibition in HepG2 cell model of steatosis. HepG2 cells were pre-treated with nSMase 2 inhibitor GW4869 (10 μM), aSMase inhibitor Imipramine (10 μM) or with nSMase 2 agonist Daunorubicin (1 μM) for 1 h, followed by overnight stimulation with oleic acid (OA; 150 μM). Activities of nSMase2 and aSMase were measured using commercial kits as described in Materials and Methods. (**A**) Changes in nSMase2 activity are shown in HepG2 cell model of steatosis. (**B**) Changes in aSMase activity are shown in HepG2 cell model of steatosis. (**C**) Changes in tricellular lipids (BODIPY 493/503). (**D**) Representative data from three independent determinations with similar results. (**E**) Representative confocal microscopy images of Nile red fluorescence staining of HepG2 cells, treated with steatosis inducing or basal (normal) conditions, obtained from three independent experiments with similar results. (**F**) *ACACA* gene expression in response to pre-treatments under steatosis condition. (**G**) *FASN* gene expression in response to pre-treatments under steatosis condition. (**H**) *CPT1α* gene expression are in response to pre-treatments under steatosis condition. (**I**) *CPT2* gene expression are in response to pre-treatments under steatosis condition. Data are expressed as mean ± SEM. * *p* ≤ 0.05, ** *p* ≤ 0.01, **** *p* ≤ 0.0001, ns: non-significant. Images are shown at 20× magnification. Scale bar = 50 μm.

**Figure 5 cells-13-00463-f005:**
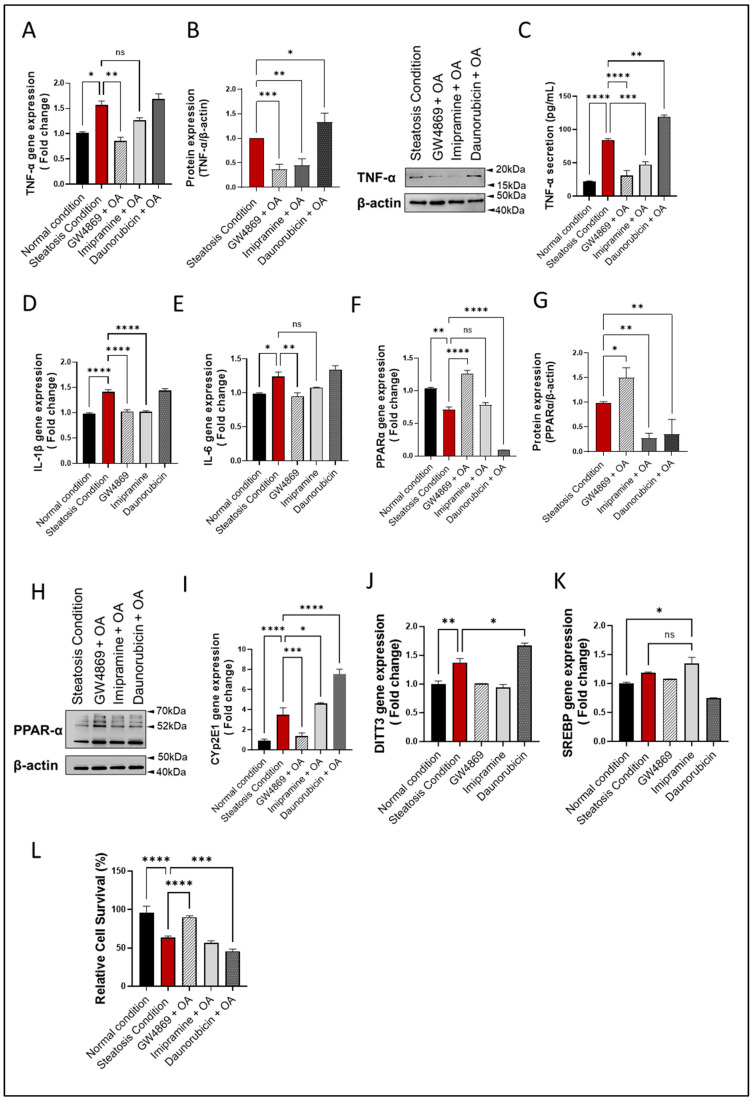
nSMase2 inhibition suppresses inflammation, oxidative stress, and apoptosis in HepG2 cell model of steatosis. HepG2 cells were exposed to steatosis-inducing vs. control conditions, with or without pre-treatments including nSMase2 inhibitor GW4869, aSMase inhibitor imipramine, and nSMase 2 agonist daunorubicin as described in Materials and Methods. Target gene expression was determined using RT-qPCR and protein expression using immunoblotting. Cell survival was determined using MTT assay. (**A**) Changes in *TNF-α* gene expression are shown in response to pre-treatments under steatosis condition. (**B**) Changes in TNF-α protein expression are shown in response to pre-treatments under steatosis condition. (**C**) TNF-α secreted protein levels in HepG2 culture supernatants are shown after pre-treatments under steatosis condition. (**D**) Changes in *IL-1β* gene expression are shown in response to pre-treatments under steatosis condition. (**E**) Changes in *IL-6* gene expression are shown in response to pre-treatments under steatosis condition. (**F**) Changes in *PPARα* gene expression are shown in response to pre-treatments under steatosis condition. (**G**) Changes in PPARα protein expression are shown in response to pre-treatments under steatosis condition. (**H**) Representative Immunoblot for PPARα protein expression. (**I**) Changes in *Cyp2E1* gene expression are shown in response to pre-treatments under steatosis condition. (**J**) Changes in *DITT3* gene expression are shown in response to pre-treatments under steatosis condition. (**K**) Changes in *SREBP* gene expression are shown in response to pre-treatments under steatosis condition. (**L**) Changes in cell viability are shown in relation to pre-treatments under steatosis condition. All data are expressed as mean ± SEM with *n* = 4. * *p*≤ 0.05, ** *p* ≤ 0.01, *** *p* ≤ 0.001, **** *p* ≤ 0.0001, ns: non-significant.

**Figure 6 cells-13-00463-f006:**
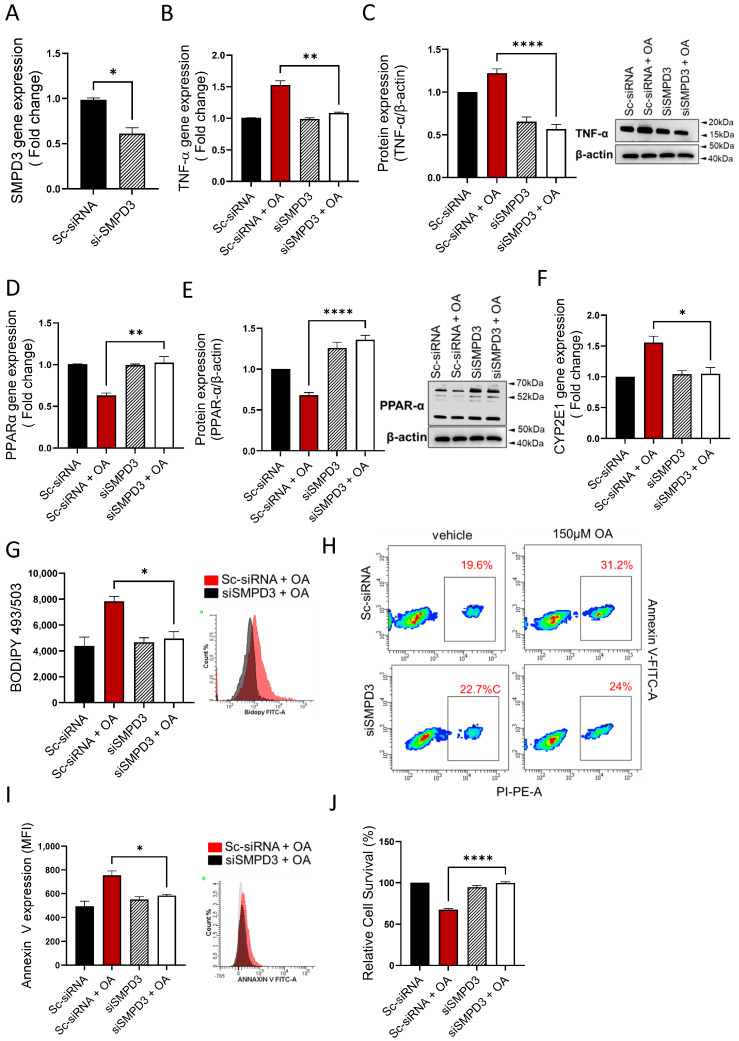
nSMase2 deficiency improves steatosis-associated pathologic changes in HepG2 cells. nSMase2 was genetically suppressed by transfecting HepG2 cells with Smpd3-specific siRNA, and controls were transfected with scrambled siRNA. Target gene expression was determined using RT-qPCR, protein expression by western blotting or flow cytometry, intracellular lipids by BODIPY 493/503 flow cytometry, apoptosis by annexin-V/PI staining, and cell viability using MTT assay as described in Materials and Methods. (**A**) Genetic suppression of Smpd3 is shown in HepG2 cells transfected with specific and scrambled siRNAs. (**B**) TNF-α gene expression is significantly reduced in Smpd3 siRNA-transfected cells under steatosis-inducing conditions, compared to controls. (**C**) TNF-α protein expression is significantly reduced in Smpd3 siRNA-transfected cells under steatosis-inducing conditions, compared to controls. (**D**) PPARα gene expression is significantly upregulated in Smpd3 siRNA-transfected HepG2 cells under steatosis-inducing conditions, compared to controls. (**E**) PPARα protein expression is significantly increased in Smpd3 siRNA-transfected cells under steatosis-inducing conditions, compared to controls. (**F**) Cyp2E1 gene expression is significantly reduced in Smpd3 siRNA-transfected HepG2 cells under steatosis-inducing conditions, compared to controls. (**G**) Representative changes in intracellular lipid accumulation, from three independent determinations with similar results, are shown. (**H**) Representative changes in apoptosis, from three independent experiments with similar results, are shown. (**I**) Bar graph of median fluorescence intensity (MFI) representing Annexin V-PI staining, from three independent determinations with similar results are shown. (**J**) Bar graph of cell viability (% survival) is shown for HepG2 cells transfected with Smpd3-specific and scrambled siRNAs. Similar data were obtained from three independent determinations. All data are expressed as mean ± SEM. * *p* ≤ 0.05, ** *p* ≤ 0.01, **** *p* ≤ 0.0001, ns: non-significant.

## Data Availability

The data that support the findings of this study are available on request from the corresponding author.

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
