# Peer review of "Neutral Sphingomyelinase 2 Inhibition Limits Hepatic Steatosis and Inflammation"

_cells, 2024, doi:10.3390/cells13050463_

Round 1

Reviewer 1 Report

Comments and Suggestions for Authors

The work by Al-Rashed et al., links elevated SMPD3 gene expression in the liver to NAFLD in HFD fed mice. The study further establishes a causal relationship by demonstrating the effects of SMPD3 gene knockdown in HepG2 cells treated with oleic acid (OA), modeling steatosis. Given the known effects of ceramides in causing ER stress, oxidative stress, insulin resistance, and inflammation, the suggestion is that inhibiting ceramide pathway genes might ameliorate these conditions.

Additionally, the reviewer has identified several areas of concern in the current study:

1. The hepatic H&E imaging of HFD-fed mice reveals only mild steatosis after 14 weeks, which seems inconsistent. Additionally, the quality of Nile Red staining is substandard. It is recommended to use Oil Red O/ORO staining and to measure hepatic triglycerides for a more comprehensive analysis.

2. To substantiate the observed phenotype, it is necessary to assess the hepatic gene expression levels of factors involved in lipogenesis, beta-oxidation, oxidative stress, ER stress, and inflammation.

3. The WBs lack molecular weight markers, raising concerns about accurately identifying the target proteins' bands.

4. OA, as an unsaturated fatty acid, is typically considered to alleviate ER stress, oxidative stress, and inflammation. Previous studies (PMID: 22961439; 23460021; 29709653; 32021639) have documented OA's mitigating effects on these stresses induced by saturated fats in various cell types and in vivo.

To conclusively demonstrate that sphingomyelins (SMs) or ceramides contribute to pathology following increased SMPD3 in the liver and in revised in vitro data, it is essential to measure the levels of SMs and ceramides both in the liver and in vitro, using appropriate fatty acid species.

The authors are advised to present their data using dot plots or box-whisker plots instead of bar graphs for clarity (PMID: 28974579).

The reviewer believes these suggestions will significantly enhance the scientific integrity and conclusions of the study.

Comments on the Quality of English Language

None.

Author Response

Response to Reviewer 1 comments.

Manuscript ID: cells-2864141  

We thank the reviewer for his thoughtful comments. We have now performed additional experiments as suggested to address the reviewer’s concerns.

Please see below point by point responses to his comments.

Comments and suggestions for authors:

The work by Al-Rashed et al., links elevated SMPD3 gene expression in the liver to NAFLD in HFD fed mice. The study further establishes a causal relationship by demonstrating the effects of SMPD3 gene knockdown in HepG2 cells treated with oleic acid (OA), modeling steatosis. Given the known effects of ceramides in causing ER stress, oxidative stress, insulin resistance, and inflammation, the suggestion is that inhibiting ceramide pathway genes might ameliorate these conditions.

Additionally, the reviewer has identified several areas of concern in the current study:

  1. The hepatic H&E imaging of HFD-fed mice reveals only mild steatosis after 14 weeks, which seems inconsistent. Additionally, the quality of Nile Red staining is substandard. It is recommended to use Oil Red O/ORO staining and to measure hepatic triglycerides for a more comprehensive analysis.

Thank you for your valuable feedback. Regarding the observation of only mild steatosis in the hepatic H&E imaging, it's important to note that all initial images were captured at 20x magnification. We recognize, in hindsight, that this magnification might have diluted the impact of the observed steatosis. To rectify this and provide a clearer, more detailed visualization, we have now included additional images at 40x and 100x magnifications for those H&Es. These higher magnification images offer a more nuanced view of the hepatic architecture and the extent of steatosis, thereby addressing the concern of inconsistency in our initial observations.

As recommended, we performed Oil Red O (ORO) staining on liver tissues alongside the original Nile Red images (revised Fig. 1H and I). Furthermore, we determined hepatic triglyceride to quantify the lipid accumulation more precisely.  Similar to Oil Red O staining, liver tissues from mice fed high fat diet for 14 weeks contained significantly higher TG (revised Fig 1L).  Please refer to revised Fig.1 for all changes implemented.

  1. To substantiate the observed phenotype, it is necessary to assess the hepatic gene expression levels of factors involved in lipogenesis, beta-oxidation, oxidative stress, ER stress, and inflammation.

Many thanks for your comment, we have conducted qRT-PCR analysis for the following genes: ACACA and FASN to investigate lipogenesis, CPT1 and CPT2 for B-Oxidation.
We have incorporated these data in (revised Fig. 4F-I) as we thought they serve the justification of the impact of lipid accumulation in the steatosis model and treatment.

3. The WBs lack molecular weight markers, raising concerns about accurately identifying the target proteins' bands.

Thank you for pointing out the absence of visible molecular weight markers in the WB images presented in our manuscript. We understand the importance of these markers for accurately identifying the target proteins' bands. To clarify, we employed Precision Plus Protein Dual Color Standards from Bio-Rad as our molecular weight marker for all WB experiments. This marker is a widely accepted standard in the field, known for its reliable and colorful display of protein sizes on membranes, facilitating the precise estimation of molecular weights. We would like to note that although the Precision Plus Protein Dual Color Standards are highly effective for visualizing protein bands, they do not directly react with the enhanced chemiluminescence (ECL) detection method used to visualize our target proteins. The ECL method specifically detects the light emitted by the labeled secondary antibody bound to the target protein, which does not interact with the colored markers. This is the reason the molecular weight markers were not visible in the ECL-developed images we submitted.

4. OA, as an unsaturated fatty acid, is typically considered to alleviate ER stress, oxidative stress, and inflammation. Previous studies (PMID: 22961439; 23460021; 29709653; 32021639) have documented OA's mitigating effects on these stresses induced by saturated fats in various cell types and in vivo.

Thank you for your comment, Oleic Acid (OA) is an unsaturated fatty acid with positive role in alleviating ER stress, oxidative stress, and inflammation. The literature you've referenced (PMID: 22961439; 23460021; 29709653; 32021639) indeed supports the view that OA can mitigate the adverse effects of saturated fats such as palmitic acid, including reducing various types of cellular stress in both cell culture and animal models. This aligns with a broader consensus on the health benefits of monounsaturated fats like OA compared to saturated fats, particularly in the context of metabolic health and inflammation.

However, its important to point out that treatment with OA is a well-recognized method for experimentally promoting the development of fatty liver in both mice and rodent models (1-2). As well as OA-induced steatosis has been studied extensively in the hepatocytes of many mammalian species, such as in HepG2 and other established cell lines (3-5). Our study builds upon this knowledge by highlighting the dual role of OA and high glucose, especially in the context of nonalcoholic fatty liver disease (NAFLD). We acknowledge OA's significance as the most abundant monounsaturated fatty acid in serum, its essential role in nutrition, and its involvement in the pathogenesis of fatty liver disease. Importantly, we also identify potential adverse effects of OA when present in excess, such as enhanced oxidative damage in rat brain and liver cells, and the induction of steatosis in liver cells. These findings underscore the complex interaction between OA and cellular health, suggesting that OA's beneficial effects in mitigating ER stress, oxidative stress, and inflammation are contingent on its concentration and the cellular context.
(1) Jiao, L., Li, H., Li, J., Bo, L., Zhang, X., Wu, W., et al. (2020). Study on Structure Characterization of Pectin from the Steamed Ginseng and the Inhibition Activity of Lipid Accumulation in Oleic Acid-Induced HepG2 Cells. Int. J. Biol. Macromol 159, 57–65. doi:10.1016/j.ijbiomac.2020.04.167
(2) i, L., Chu, X., Yao, Y., Cao, J., Li, Q., and Ma, H. (2020). (-)-Hydroxycitric Acid Alleviates Oleic Acid-Induced Steatosis, Oxidative Stress, and Inflammation in Primary Chicken Hepatocytes by Regulating AMP-Activated Protein Kinase-Mediated Reactive Oxygen Species Levels. J. Agric. Food Chem. 68, 11229–11241. doi:10.1021/acs.jafc.0c04648
(3) Alkhatatbeh, Mohammad J., Lisa F. Lincz, and Rick F. Thorne. "Low simvastatin concentrations reduce oleic acid-induced steatosis in HepG2 cells: An in vitro model of non-alcoholic fatty liver disease." Experimental and Therapeutic Medicine 11.4 (2016): 1487-1492.
(4) J. Hu, W. Hong, K. Yao, X. Zhu, Z. Chen, L. Ye Ursodeoxycholic acid ameliorates hepatic lipid metabolism in LO2 cells by regulating the AKT/mTOR/SREBP-1 signaling pathway
(5) Song, Huiqi, et al. "Oleic acid-induced steatosis model establishment in LMH cells and its effect on lipid metabolism." Poultry Science 102.1 (2023): 102297.

To conclusively demonstrate that sphingomyelins (SMs) or ceramides contribute to pathology following increased SMPD3 in the liver and in revised in vitro data, it is essential to measure the levels of SMs and ceramides both in the liver and in vitro, using appropriate fatty acid species.

Thanks for the comment. We agree that measuring levels of SMs and ceramides would have further strengthened the findings of this study but, unfortunately, this experiment could not be done due to reagents unavailability. Surely, we will do such experiments in our follow-up studies.

The authors are advised to present their data using dot plots or box-whisker plots instead of bar graphs for clarity (PMID: 28974579).

Thanks for the comment. In addressing this statement, all figure legends have been accordingly modified to clarify the number of replicates of each experiment (n), as well as the number of independent experiments conducted. Please note that we have changed the presentation of our bar graphs in Figure 1 to box-whisker plots. The reviewer believes these suggestions will significantly enhance the scientific integrity and conclusions of the study.
Thank you for your insightful suggestions. We greatly appreciate your thorough review and believe that incorporating your recommendations will indeed strengthen the scientific integrity and the conclusiveness of our study.

Reviewer 2 Report

Comments and Suggestions for Authors

The only strength of this study is that the authors identified a novel target. However, in the case of cell lines, the metabolism is significantly different from that of primary cells, and it is difficult to expect a clear therapeutic effect without in vivo experiments. In some experiments, there is no proper control group (Fig. 5B and E). The beta-actin bands also appear to be identical (Fig. 6C and E). Moreover, the overall human relevance of the current study remains unclear.

Author Response

Response to Reviewer 2 comments.

Manuscript ID: cells-2864141  

We thank the reviewer for his thoughtful comments. We have now performed additional experiments as suggested to address the reviewer’s concerns.

Please see below point by point responses to his comments.

The only strength of this study is that the authors identified a novel target.

However, in the case of cell lines, the metabolism is significantly different from that of primary cells, and it is difficult to expect a clear therapeutic effect without in vivo experiments.

Many thanks for your comment, we concur with your observation regarding the distinct metabolic profiles of cell lines compared to primary cells, and we wish to clarify that our in vitro findings are not intended to be directly equated with in vivo outcomes. As noted in our manuscript (lines 642-644), we acknowledge the necessity for comprehensive in vivo testing to validate the therapeutic effects observed. Nonetheless, it's critical to highlight that, despite its simplicity, our in vitro model serves as a valuable tool for preliminary drug efficacy screening and mechanistical impact. This approach fills a significant gap in current research methodologies, offering an initial yet crucial step towards identifying potential therapeutic candidates.

In some experiments, there is no proper control group (Fig. 5B and E).

Thank you for your comment, in response to your observation concerning the lack of a proper control group in certain experiments (Fig. 5B and E), we wish to emphasize that the primary objective of our study was to elucidate the effects of SM modulation on predicted steatosis. While it is true that Figures 5B and E do not include a conventional control group, this was a deliberate methodological choice. The outcomes and interpretations drawn from these specific figures are supported by comprehensive control data presented in earlier sections of our work. These control setups were designed to robustly establish baseline conditions against which the impact of SM modulation on steatosis could be accurately assessed. We believe that the cumulative evidence presented throughout our study adequately supports our conclusions, despite the unconventional presentation of control data in the figures mentioned.

The beta-actin bands also appear to be identical (Fig. 6C and E). Moreover, the overall human relevance of the current study remains unclear.

Many thanks for your comment, in addressing your concerns regarding the apparent identical nature of the beta-actin bands in Figures 6C and E, we appreciate the opportunity to clarify that the proteins under investigation have distinct molecular sizes; TNF-a: 17kDa, PPARa : 50kDa while the B-actin : 42 kDa  . This characteristic enabled us to run them on the same membrane, thus ensuring that the beta-actin served as a consistent loading control across different lanes. As for the human relevance, we will bring back the same argument of the limitation of our study and the requirement of further investigations to understand this observation in NAFLD patients.  

Reviewer 3 Report

Comments and Suggestions for Authors

In the manuscript entitled “Neutral Sphingomyelinase 2 Inhibition Limits Hepatic Steatosis and Inflammation”, the author observes an upregulation of the gene encoding neutral sphingomyelinase SMPD3 in the liver tissues of mice with liver steatosis. They demonstrate that both pharmacological and genetic inhibition of nSMAse2 can prevent intracellular lipid accumulation and inflammation in a HepG2-steatosis cellular model. The finding of this manuscript is interesting. However, the paper suffers from lots of problems that detract from the overall quality, and I have some comments shown below to improve the clarity of this manuscript.

  1.     Given that most conclusions drawn in this manuscript are based on findings from in vitro systems, it would be appropriate to modify the title to “Neutral Sphingomyelinase 2 Inhibition Limits Hepatic Steatosis and Inflammation in vitro”. 2.    It would be better to include references about the role of ceramides in NAFLD. 3.    There appears to be an inconsistency between the method described for the liver steatosis mouse model in lines 83 to 88 and the description provided in the Abstract (HFD with high sucrose). 4.    Liver Function Tests (ALT and AST assay) should be included to get the conclusion that “High-fat diet (HFD) feeding induces chronic liver injury and a NAFLD phenotype in mice” in Figure 1. 5.    For Figure 4, GW4869, Imipramine, or Daunorubicin without OA treatment should be used to assess their effects on basal nSMase2 activity. The lipid accumulation shown in Figure 4D is inconsistent with that shown in Figure 4C, please justify. 6.    The images presented in Figure 6 appear inconsistent with those described in the Figure legend and manuscript.  

Author Response

Response to Reviewer 3 comments.

Manuscript ID: cells-2864141  

We thank the reviewer for his thoughtful comments. We have now performed additional experiments as suggested to address the reviewer’s concerns.

Please see below point by point responses to his comments.

In the manuscript entitled “Neutral Sphingomyelinase 2 Inhibition Limits Hepatic Steatosis and Inflammation”, the author observes an upregulation of the gene encoding neutral sphingomyelinase SMPD3 in the liver tissues of mice with liver steatosis. They demonstrate that both pharmacological and genetic inhibition of nSMAse2 can prevent intracellular lipid accumulation and inflammation in a HepG2-steatosis cellular model. The finding of this manuscript is interesting. However, the paper suffers from lots of problems that detract from the overall quality, and I have some comments shown below to improve the clarity of this manuscript.

  1.     Given that most conclusions drawn in this manuscript are based on findings from in vitro systems, it would be appropriate to modify the title to “Neutral Sphingomyelinase 2 Inhibition Limits Hepatic Steatosis and Inflammation in vitro”.

Thank you for your insightful recommendation. We revised the title as suggested to read: " Neutral Sphingomyelinase 2 Inhibition Limits Hepatic Steatosis and Inflammation in vitro".

 2.    It would be better to include references about the role of ceramides in NAFLD.

Many thanks for your comment, we have edited our introduction to include more references regarding the role of ceramide in NAFLD the amended part reads as follows:

Numerous studies have implicated ceramide dysregulation in the pathogenesis of non-alcoholic fatty liver disease (NAFLD) [1, 2 ]. Ceramides have been shown to contribute to insulin resistance, inflammation, and lipid accumulation in hepatocytes, key features of NAFLD [3]. And believed to modulate cellular and whole-body metabolism, potentially exacerbating NAFLD progression [4 ]. Regardless, this role of ceramides is believed to be multifaceted when it comes to hepatocyte death and liver injury. The de novo pathway of ceramide synthesis, in particular, has been extensively studied and im-plicated in contributing to liver damage in conditions such as NAFLD. However, our un-derstanding of other pathways, such as the sphingomyelin pathway, and their specific roles in liver injury remains limited.

New references added :

  1. Hannun YA, Obeid LM. Principles of bioactive lipid signalling: lessons from sphingolipids. Nat Rev Mol Cell Biol. 2008;9(2):139-150. doi:10.1038/nrm2329
  2. Chavez JA, Summers SA. A ceramide-centric view of insulin resistance. Cell Metab. 2012;15(5):585-594. doi:10.1016/j.cmet.2012.04.002
  3. Holland WL, Summers SA. Sphingolipids, insulin resistance, and metabolic disease: new insights from in vivo manipulation of sphingolipid metabolism. Endocr Rev. 2008;29(4):381-402. doi:10.1210/er.2007-0025
  4. Bikman BT, Summers SA. Ceramides as modulators of cellular and whole-body metabolism. J Clin Invest. 2011;121(11):4222-4230. doi:10.1172/JCI57144

 3.    There appears to be an inconsistency between the method described for the liver steatosis mouse model in lines 83 to 88 and the description provided in the Abstract (HFD with high sucrose).

Thank you for your comment, the inconsistency arises from the fact that the standard high-fat diet (HFD) used in our study is indeed high in sucrose. We apologize for any confusion caused. To provide clarity, we want to specify that the HFD utilized in our study (Cat number D12331i from Research Diets Inc.) is the "Western diet" or "Western-style diet," which is commonly characterized by its high fat content along with added sugars such as sucrose. This diet is often referred to as the "high-fat, high-sucrose diet" or "high-fat, high-sugar diet."

It's important to note that there are variations of HFDs, including those high in fat but low in sugar, commonly referred to as "high-fat, low-carbohydrate diets." However, for our experimental purposes, we utilized the standard HFD with high sucrose content. To prevent this confusion we have amended our methodology as follow “ To induce steatosis by HFD feeding, mice (n = 5) were fed "Western-style diet," which is commonly characterized by its high fat content along with added sugars such as sucrose, with 58%  of the calories coming from fat” 

 4.    Liver Function Tests (ALT and AST assay) should be included to get the conclusion that “High-fat diet (HFD) feeding induces chronic liver injury and a NAFLD phenotype in mice” in Figure 1. 5.   

Thank you for your suggestion regarding the inclusion of Liver Function Tests (ALT and AST assay) in our study. We appreciate your insight into enhancing the comprehensiveness of our research. As recommended, we have performed liver ALT and AST function assay’s for this particular study. However, no significant difference was seen between CD and HFD group. We have included this data as supplementary file  (Supplementary Fig. S1A and B).

For Figure 4, GW4869, Imipramine, or Daunorubicin without OA treatment should be used to assess their effects on basal nSMase2 activity.

Thank you for your comment. In Figure 4A and B present the basel nSMase2 activity without OA treatment.

5. The lipid accumulation shown in Figure 4D is inconsistent with that shown in Figure 4C, please justify.

Thank you for your observation regarding the lipid accumulation depicted in Figures 4C and 4D.
Figure 4C illustrates the mean fluorescence intensity (MFI) of BODIPY 493/503, which was quantified using flow cytometry. The subsequent representative histograms provide a visual representation of the data. In contrast, Figure 4D presents representative images of Nile Red immunofluorescence staining. We believe that the quantitative ability of flow cytometry provides a more accurate and precise measurement of lipid accumulation compared to the semi-quantitative protocol used in immunofluorescence staining. Therefore, any perceived inconsistencies between the two figures may be attributed to the differences in the techniques employed for lipid assessment.

6. The images presented in Figure 6 appear inconsistent with those described in the Figure legend and manuscript.
Thank you for bringing this to our attention. We have thoroughly reviewed all the figures, their legends, and the corresponding descriptions in the manuscript to ensure accuracy and consistency. Upon careful examination, we have confirmed that the images presented in Figure 6 are accurately described in both the legend and the manuscript.

Round 2

Reviewer 1 Report

Comments and Suggestions for Authors

The reviewer appreciates author's efforts in revising and enhancing the manuscript. Upon further examination, the reviewer has identified few issues (Point 4 and 5) that were not addressed in the initial review. Regrettably, the reviewer apologize for the oversight in not identifying these point during the first round of revisions. Further, the reviewer would also like to highlight unaddressed comments from the first revision.  

1. The inclusion of the Neil Red images/measurements within the study is advised against due to their subpar quality. It would be beneficial for the authors to omit these from their study entirely. ORO measurements are far much clear and similar. 

2. The authors have provided a commendable explanation on the theoretical significance of the Precision Plus Protein Dual Color Standards from Bio-Rad. The reviewer acknowledges and appreciates this effort. However, there is ambiguity concerning the PPARa blots; specifically, it is unclear whether the upper band truly represents PPARa or if it is the lower bands that do. To resolve this, the authors must clearly indicate the positions of the actual molecular weight markers relative to the film on each blot presented across the various WBs. This is crucial as misinterpretation of bands is a common error, and the absence of clearly indicated markers exacerbates this issue.

3. It is recommended that the authors supplement all bar graphs with dot plots to enhance the clarity and interpretability of the data presented.

4. The manuscript fails to detail the expression of SMPD3 under various conditions (as illustrated in Figure 6). Given that various treatments may influence transfection efficiency in cells, it is imperative for the authors to address this oversight.

5. Regarding Figure 2, the clarity of the figure is lacking, making it difficult to understand and interpret. Furthermore, the rationale behind selecting SMPD3 from the transcriptomic analysis remains underexplained. The panel H data indicate other genes that are upregulated more significantly than SMPD3. The authors are encouraged to elucidate why these genes were not selected for targeting, especially considering that the targeting of the ceramide pathway is not a novel approach within NAFLD pathology research. Numerous studies have illustrated the benefits of targeting ceramides in ameliorating NAFLD. Thus, the authors should provide a compelling justification for their study’s focus and how it distinguishes itself from existing research.

Author Response

Response to Reviewer 1 comments.

Manuscript ID: cells-2864141  

Comments and suggestions for authors:

The reviewer appreciates author's efforts in revising and enhancing the manuscript. Upon further examination, the reviewer has identified few issues (Point 4 and 5) that were not addressed in the initial review. Regrettably, the reviewer apologize for the oversight in not identifying these point during the first round of revisions. Further, the reviewer would also like to highlight unaddressed comments from the first revision.  

  1. The inclusion of the Neil Red images/measurements within the study is advised against due to their subpar quality. It would be beneficial for the authors to omit these from their study entirely. ORO measurements are far much clear and similar. 

Many thanks for the suggestion. We have removed all Nile Red staining data from Fig.1 as advised. Fig. 1 was revised as follow:

  1. The authors have provided a commendable explanation on the theoretical significance of the Precision Plus Protein Dual Color Standards from Bio-Rad. The reviewer acknowledges and appreciates this effort. However, there is ambiguity concerning the PPARa blots; specifically, it is unclear whether the upper band truly represents PPARa or if it is the lower bands that do. To resolve this, the authors must clearly indicate the positions of the actual molecular weight markers relative to the film on each blot presented across the various WBs. This is crucial as misinterpretation of bands is a common error, and the absence of clearly indicated markers exacerbates this issue.

Thank you for your constructive feedback. We incorporated clear annotations indicating the positions of the actual molecular weight markers relative to the film for each blot presented.

  1. It is recommended that the authors supplement all bar graphs with dot plots to enhance the clarity and interpretability of the data presented.

Thank you for your insightful suggestion regarding the presentation of our data with both bar graphs and dot plots. We are pleased to implement dot plots for our animal study data as we implemented the whisker box, where the variability and individual data points offer significant interpretative value. However, for our cell culture data, due to the highly controlled experimental conditions that minimize variability, dot plots may not provide additional representational benefit.

  1. The manuscript fails to detail the expression of SMPD3 under various conditions (as illustrated in Figure 6). Given that various treatments may influence transfection efficiency in cells, it is imperative for the authors to address this oversight.

We appreciate the reviewer's attention to the details regarding SMPD3 expression as presented in Figure 6. The data presented in figure 6 highlights the impact of SMPD3 deficiency through the silencing of SMPD3 gene.  The experimental model took in consideration all conditions (which are the presence and absence of steatosis trigger that consist of high glucose / Oleic acid) to elucidate the role of nSMase2 in promoting liver steatosis. Our experimental design compared the effects of siSMPD3 in both the presence and absence of a steatosis model trigger against a siRNA vehicle control, under similar conditions. This comprehensive approach was intended to cover all necessary conditions to understand the impact of SMPD3 deficiency.

  1. Regarding Figure 2, the clarity of the figure is lacking, making it difficult to understand and interpret. Furthermore, the rationale behind selecting SMPD3 from the transcriptomic analysis remains underexplained. The panel H data indicate other genes that are upregulated more significantly than SMPD3. The authors are encouraged to elucidate why these genes were not selected for targeting, especially considering that the targeting of the ceramide pathway is not a novel approach within NAFLD pathology research. Numerous studies have illustrated the benefits of targeting ceramides in ameliorating NAFLD. Thus, the authors should provide a compelling justification for their study’s focus and how it distinguishes itself from existing research.

Thank you for your feedback on Figure 2 and the questions regarding our selection of SMPD3 for further analysis. The clarity of Figure 2 will be improved in our revision to facilitate better understanding and interpretation. Our selection of SMPD3 from the transcriptomic analysis was guided by its specific upregulation pattern and its known role in ceramide metabolism, which is pivotal in NAFLD progression. While other genes showed higher upregulation, SMPD3 was chosen due to its direct relevance to the ceramide pathway, a critical target in NAFLD research. Our study aims to build on existing knowledge by providing novel insights into SMPD3's specific contributions within the broader context of NAFLD pathology under high fat feeding, thereby offering a distinct perspective from previous research.

Reviewer 2 Report

Comments and Suggestions for Authors

All issues are addressed for publication in this journal.

Author Response

We are grateful for your thorough evaluation and acceptance of our manuscript.
